# Electrospun PA66/Graphene Fiber Films and Application on Flexible Triboelectric Nanogenerators

**DOI:** 10.3390/ma15155191

**Published:** 2022-07-26

**Authors:** Qiupeng Wu, Zhiheng Yu, Fengli Huang, Jinmei Gu

**Affiliations:** 1College of Mechanical Engineering, Zhejiang University of Technology, Hangzhou 310014, China; 2112002087@zjut.edu.cn; 2Key Laboratory of Advanced Manufacturing Technology of Jiaxing City, Jiaxing University, Jiaxing 341000, China; jmgu1218@zjxu.edu.cn; 3College of Mechanical and Electrical Engineering, Jiaxing Nanhu University, Jiaxing 314000, China; yuzhiheng@jxnhu.edu.cn

**Keywords:** triboelectric nanogenerators, PA66/graphene fiber films, PVDF films, performance, wearing comfort

## Abstract

Triboelectric nanogenerators (TENGs) are considered to be the most promising energy supply equipment for wearable devices, due to their excellent portability and good mechanical properties. Nevertheless, low power generation efficiency, high fabrication difficulty, and poor wearability hinder their application in the wearable field. In this work, PA66/graphene fiber films with 0, 1 wt%, 1.5 wt%, 2 wt%, 2.5 wt% graphene and PVDF films were prepared by electrospinning. Meanwhile, TENGs were prepared with PA66/graphene fiber films, PVDF films and plain weave conductive cloth, which were used as the positive friction layer, negative friction layer and the flexible substrate, respectively. The results demonstrated that TENGs prepared by PA66/graphene fiber films with 2 wt% grapheme showed the best performance, and that the maximum open circuit voltage and short circuit current of TENGs could reach 180 V and 7.8 μA, respectively, and that the power density was 2.67 W/m^2^ when the external load was 113 MΩ. This is why the PA66/graphene film produced a more subtle secondary network with the addition of graphene, used as a charge capture site to increase its surface charge. Additionally, all the layered structures of TENGs were composed of breathable electrospun films and plain conductive cloth, with water vapor transmittance (WVT) of 9.6 Kgm^−2^d^−1^, reflecting excellent wearing comfort. The study showed that TENGs, based on all electrospinning, have great potential in the field of wearable energy supply devices.

## 1. Introduction

With the development of Internet of Things technology (IoT), portable, flexible wearable electronic devices with various functions have been widely considered. In the IoT society, various devices offer many possibilities through data transmission and processing [1,2,3,4], such as real-time health monitoring, environmental monitoring, as well as biomedical assistance and human–machine interfaces [5,6,7,8,9]. It has brought about great changes in our lives. However, the flexibility and miniaturization of these electronic devices put forward higher requirements for power supply [10,11,12]. Triboelectric nanogenerators (TENGs), based on the combined action of friction generation and electrostatic induction, convert mechanical energy wasted in daily life into electrical energy required to drive tiny flexible electronic devices [13,14,15]. It has the advantages of high efficiency, wearable and environmental protection and suitability for integration into many flexible electronic devices.

All electrospun nanofiber films are widely used in TENGs design. The electrospun fibrous films have remarkable properties, such as inherent rough structure, large specific surface area and hierarchical porous structure [16,17], which could effectively increase the friction performance and air permeability, and the introduction of functional materials by electrospinning could affect the chemical and electrical properties of the resulting composites [18], optimizing the overall performance and TENGs’ durability. Based on the previous studies, a series of effective microstructured electrospun films was prepared using different materials, such as polyvinylidene fluoride (PVDF) [19,20]. Meng [21] et al. successfully fabricated TENGs based on an all-electrospun silk fibroin/carbon nanotube (CNT) film as a substrate, with a power density of 317.4 μW/cm^2^ under hand patching. Incorporating functional materials into polymer nanofibers could improve electrical output performance. Currently, most research focuses on improving output performance, such as selection of friction materials [22,23,24], surface modification [25,26,27] and micropatterning [28,29,30]. However, the above experiments all used dense metal films as electrodes. When worn, the breathability is reduced, making TENGs deficient in ideal breathability and wearing comfort, so that TENGs are short of the ideal air permeability and wearing comfort. Additionally, previous studies focused on materials or complex structures to a certain extent, ignoring the comfort of wearing. In fact, the multi-scale micro-nano structures in the friction layer structure could not only improve the output performance of TENGs, but also improve their mechanical properties and wearing comfort.

In this work, a method for all-electrospun fiber-based TENGs, consisting of two parts, including a PA66/graphene nanofiber-film-rubbed positive base layer and PVDF-rubbed negative electrode layer, was proposed. On the one hand, a PA66/graphene nanofiber film with multi-layer micro-nano structure could not only increase the effective friction area of contact, but also graphene dispersed in the PA66 fiber could be used as a charge capture site, which could increase the surface charge on the film surface, greatly improving its triboelectric properties [19]. In addition, all functional layers of the nanofiber-film-based triboelectric nanogenerators (TENGs) were constructed from nanofiber networks with a layered structure, ensuring air and moisture permeability. TENGs exhibited good electrical output performance as well as excellent flexibility and wear resistance. The maximum output voltage and current of the TENGs fabricated could reach 180 V and 7.8 μA, respectively, corresponding to a power density of 2.67 W/m^2^. Water vapor transmission rate could be up to 9.6 Kgm^−2^d^−1^. In addition, the TENGs prepared could be placed anywhere on the body, detecting various human movements, and the mechanical energy that is wasted in life could be obtained. This energy can instead be used to power wearable electronic devices in daily life.

## 2. Materials and Methods

### 2.1. Materials

Polyamide66 (PA66) was bought from Aladdin Pharmaceuticals (Shanghai, China). Graphene was sold by Shenzhen Turing Artificial Intelligence Black Technology Co., Ltd. (Shenzhen, China). Polyvinylidene Difluoride (Mw 600000, PVDF) was produced by Arkema (Paris, France). Formic Acid and Polyethylene Oxide (PEO) were bought from Macklin (Shanghai, China). N, N-Dimethylformamide (DMF) and Acetone were produced by Sinopharm (Beijing, China). Plain weave conductive fabric was produced by Saintyear Electronic (125 ± 25 um, Hangzhou, China).

### 2.2. Fabrication of PA66/Graphene Fiber Film Layer

Firstly, 0.5 wt% PEO was added to 10 mL of Formic Acid and a magnetic stirrer was used to stir for 1 h to increase the viscosity of the solution, so that it could be well electrospun. Secondly, 15 wt% PA66 particles were put into Formic Acid/PEO solution, which were fully dissolved via a magnetic stirrer for 4 h to obtain 15 wt% PA66 solutions. Finally, different mass fractions of graphene (1 wt%, 1.5 wt%, 2 wt%, 2.5 wt%) were put into the prepared PA66 solution, respectively, and stirred to fully mix. Further, ultrasonic dispersion was applied for 6 h to make the graphene evenly dispersed in the solution.

The electrospinning process was performed with a supply voltage of 20 kV, a working distance between the spray nozzle and the substrate was 15 cm and the PA66/graphene solution was injected into a 10 mL syringe with a blunt needle (0.6 mm inner diameter). The injection rate of the syringe was 0.1 mL/h. PA66/graphene nanofiber films were collected on plain weave conductive cloth (Ni-Cu) fabric electrodes, which were prepared at 25 °C and relative humidity of 50 ± 5%. The thickness of PA66/graphene fiber film was 80 ± 10 μm, measured by Talysurf Profiler (Dektak-XT-10 th, Bruker, MA, USA).

### 2.3. Fabrication of PVDF Nanofiber Films

Firstly, 3 mL Acetone was added into 7 mL DMF to obtain a DMF/Acetone solution in a 7-to-3 volume ratio. Secondly, 10 wt% PVDF powder was added into the mixed solution and magnetically stirred for 4 h under heating in a 60 °C water bath. Finally, the 10 wt% clear and transparent PVDF solution was obtained.

The electrospinning process was performed with a supply voltage of 12 kV, while the working distance between the spray nozzle and the substrate was 10 cm. The advance rate of the syringe was 0.4 mL/h. The PVDF nanofiber membrane was collected on the conductive fabric electrode, which was also collected under the same conditions, mentioned in Section 2.2. However, when dried under vacuum, the drying time of the PVDF was 80 °C for 4 h, 60 °C for 3 h and 50 °C for 1 h based on the previous experiment. The thickness of PVDF nanofiber films was 130 ± 15 um.

### 2.4. Assembly of TENG-Based Nanofiber Films

Based on the electrospinning process, the composite fabric of the two friction layers of TENGs was prepared. Firstly, the double-sided conductive tape was used as the electrode lead, which was tightly pasted on the back of the conductive fabric substrate where the different films were located. Then, the required size could be trimmed, such as 4 cm × 4 cm. Finally, these breathable films could be pasted on the corresponding positions of daily clothes, collecting the mechanical energy generated by wasted human movement.

### 2.5. Measurement of the TENGs

The surface morphologies of PA66 films, PA66/graphene films and PVDF films were characterized by the Field Emission Scanning Electron Microscopy (Hitachi S-4800, Tokyo, Japan). X-ray Diffraction (XRD, HAOYUAN DX-2700BH, Dandong, China) was used to study the elements and material structure of the electrospun fiber films. The I_SC_ and V_OC_ were measured by Keithley System Electrometer (6517b, Tektronix, Shanghai, China).

## 3. Results and Discussion

### 3.1. Preparation of the TENGs

The main concept of the design in this work was to use common fabric materials as the basic building unit to construct a wearable TENGs platform, so as to achieve effective harvesting of low-frequency, ubiquitous and easily wasted human biomechanical energy, such as walking and running. The all-electrospinning-based power-generating textile had multiple secondary structures and it could be better integrated with personal clothing, powering wearable electronic devices. These power-generating fabrics with excellent mechanical properties exhibited excellent breathability and cutability. It was an ideal device to make energy harvesters and sensors of a flexible wearable electronic system integrated, which guaranteed the integration of TENGs and wearable electronic devices with clothing.

A detailed preparation process for the PA66/graphene nanofiber film and PVDF film are schematically illustrated in Figure 1. The prepared organic solution flowed out slowly at a constant speed under the action of a syringe pump. Under the action of tens of thousands of volts of electrostatic field, the droplet at the needle of the organic solution could form a Taylor cone and the solvent in the sprayed droplet could evaporate in the air, becoming nano-scale filaments [31]. Finally, it was collected by the cylinder connected to the negative pole.

The prepared TENGs were composed of three functional components, including a PA66/graphene fiber film and PVDF film with two tribological polarities for contact electrification, and a plain weave conductive cloth for charge conduction. Based on previous studies [30], constructing a nanostructured triboelectric film with high-effective surface area was an effective method to improve the friction performance of TENGs. Electrospinning was widely used to fabricate fibrous films with surface structures. The PA66/graphene and PVDF films prepared by electrospinning were both thin and flexible. Due to the addition of graphene, the film was gray black, shown in Figure 2a. With the addition of graphene, the conductivity in the spinning solution increases; meanwhile, the viscosity in the spinning solution also increases. This results in an increase in the conductivity of the solution and the charge density, leading to reducing jet stability during spinning. This promotes fiber splitting, which results in the formation of a secondary fiber network (Figure 2b).The increase in solution viscosity could inhibit this phenomenon, so when too much graphene is added, the solution viscosity will be too high and the secondary network will be reduced [31].The generation of the secondary network can make the two friction layers have more contact area when they are in contact, promoting more charge flow when the TENGs works and improving the output performance. On the other hand, graphene dispersed between nanofibers could serve as charge-trapping sites, which could store more charge during the contact separation process of TENGs, enhancing the triboelectric surface charge and increasing the triboelectric properties of TENGs. For prepared TENGs, plain weave conductive cloth used instead of metal electrodes, shown in Figure 2d, greatly enhancing the air permeability and wearing comfort of electronic devices.

### 3.2. Effects of Graphene with Different Mass Fractions on Output Performance

To study the elements in the films, the XRD patterns of PA66/graphene thin-film layers with different mass fractions of graphene were observed, shown in Figure 3. The two peaks (2θ = 19° and 2θ = 23.5°) correspond to the characteristic diffraction peak of an α-crystal form and were, therefore, obtained. What is more, the crystal form of the substance in the PA66 film was consistent with that of the PA66 particles. Small peaks around 45 and 50 degrees appeared and this is the reason that the PA66/graphene with different graphene mass fractions was fixed pinned with the adhesive tape, leading to the introduction of impurities. The results showed that the addition of graphene could not change the peak position of PA66 and from the diffraction peaks of the XRD pattern, it can be determined that the crystalline form of the material in the PA66 film is consistent with the PA66 particles, so it can be concluded that the material structure of the PA66 material in the film prepared by spinning has not changed.

To study the effect of graphene doping on the output performance, contact-separation-mode TENGs based on PA66/graphene and PVDF films were designed. The two films were glued to two 3D-printed slabs with bosses (Figure 4a,b). The upper and bottom plates are connected by four studs. Between the two plates, four springs are placed over the studs to provide resilience (Figure 4c). This device can make the two friction layers contact under the slap and quickly return to the original position when the external force is removed. At the same time, the external force on contact can be controlled by replacing springs with different elastic coefficients. Connecting the lead-out conductive tape to the Keithley 6517b test leads (Figure 4d), the resulting voltage and current signals were collected by the Keithley 6517b for analysis. The synergistic effect of contact charging and electrostatic induction is shown in Figure 5a, which laid the theoretical foundation for TENGs as an energy harvester [32]. The triboelectric pair started with a separation attitude; no charge was generated at this time and there was no potential difference between the two triboelectric layers. When the TENGs were stimulated by an external force, the two triboelectric layers approached each other until they were in contact, and the two nano-friction layers would slide relative to each other at the microscopic level, due to the different electron affinities in the two friction materials, and the two friction layers would generate opposite charges. When the two friction layers were separated, a potential difference would be formed between the two friction layers to promote the flow of free electrons from the PVDF film to the PA66/graphene film through the external load, generating an instantaneous current. As the external force was continuously unloaded, this potential difference would continue to increase, and when it was fully restored to its original position, the potential difference would reach its maximum value. As the external force approached the PA66/graphene film again, the original electrostatic balance was broken again, causing the electrons in the PA66/graphene film to run back to the PVDF film, resulting in an opposite instantaneous current. When the two sides were in full contact, the potential difference became zero again. This continuous contact and separation process generated pulses of voltage and currents of opposite polarity, converting mechanical energy into electrical energy.

According to previous studies [33,34,35], when the change in the distance between the two friction layers was much smaller than the area of the friction layer, the open-circuit voltage and short-circuit current of the TENGs could be calculated as follows:(1)$$V_{OC}^{TENG}=\frac{Qsc(x)}{C(x)}=\frac{\sigma x(t)}{\varepsilon_{0}}$$
(2)$$I_{SC\;\max}^{TENG}=\frac{S\sigma v(t)}{d_{0}}=\frac{2\pi S\sigma fx_{\max}}{d_{0}}$$
where *C*(*x*) was the capacitance between the two electrodes at different time-varying displacements *x*; *ε*_0_ was the vacuum dielectric constant; *S* was the surface area of the friction layer; *d*_0_ was the effective thickness of the dielectric layer; *σ* was the triboelectric surface charge density; *Q_SC_*(*x*) was the amount of short-circuit charge transfer at displacement *x*; and *f* was the frequency. Therefore, according to Equations (1) and (2), we could observe that the open-circuit voltage of TENGs was independent of frequency *f* and increased only with larger displacement between the two contact surfaces, and the short-circuit current was proportional to the frequency *f**,* so the measured maximum short-circuit current increased with the frequency *f*, shown in Figure 5b,c. The results showed that TENGs still had good output performance at low frequencies and could collect low-frequency energy. Compared to the short-circuit current (*I_SC_*) and open-circuit voltage (*V_OC_*) of the TENGs and an external force exerted with a frequency of 4 Hz, a higher circuit signal and output power could be obtained.

### 3.3. Effects of the Dielectric Constant of the Electrospun Film on Output Performance

With the increase in graphene, the dielectric constant of the electrospun film also had an effect on the output performance. The enhanced dielectric constant of the triboelectric layer could improve the ability to transfer charge density and the surface potential also had a large impact on the performance of TENGs. A higher dielectric constant could use the electrospun PA66/graphene film as a friction layer to store more charge. However, an excessively large dielectric constant would generate a larger dielectric loss, which leads to the leakage of charge. Due to the changes in frequency and external strain during human motion, it was necessary to explore their effects on the output performance of wearable devices. As shown in Figure 3, the PA66/graphene fiber films of graphene with different mass fractions were used as the positive friction layer and PVDF was tested as the TENGs of the friction negative layer (both applied 4 Hz, 30 N external force), shown in Figure 5d,e. In the range of graphene mass fraction of 0–2 wt%, the short-circuit current and open-circuit voltage signals of TENGs increased with the increase in graphene mass fraction, while the maximum value could reach 180 V and 7.8 μA, respectively, when the graphene content was 2 wt%. Compared with the fiber film without graphene added, the modified *V_OC_* and *I_SC_* were both increased by about four-times. Therefore, the TENGs made of PA66/graphene fiber film with 2 wt% graphene content were the most suitable experimental sample. Further, the piezoelectric properties of PVDF films were tested by the method reported by Yin [36], so the highest output was only 0.8V, meaning, in this paper, the piezoelectric effect of PVDF films was not considered.

### 3.4. Effects of Load Resistance on Output of Voltage and Current

To study the effects of load resistance on output of the voltage and current, the output performance of nanofiber-film-based TENGs was researched, shown in Figure 6. As the load resistance increased, the voltage tended to increase, while the current tended to decrease due to ohmic losses. The output current and output voltage peak values could be reached 180 V and 7 μA, respectively. According to the formula *P* = *I*^2^*R*, the power of TENGs could be calculated. At 113 MΩ resistance, the maximum power density could reach 2.67 W/m^2^. The generated output power was sufficient to drive some low-energy electronic devices, solving the problem of sustainable power supply for wearable systems in a stable and reliable manner effectively.

Due to the striking features of the porous structure, the fabricated power generation fabric had excellent water vapor permeability, which can be visually seen in Figure 7a. As we can see, the composite fabric was covered over a beaker filled with hot water. The abundant water vapour could move freely through the fabric without any restraint. To quantitatively demonstrate the breathability, the fabric was tested for water vapor transmission rate (WVTR) according to the ASTM E96 inverted cup standard at a relative humidity of 50% and a temperature of 38 °C. The WVTR value was calculated as follows.
(3)$$WVT\;\mathrm{rate}=\frac{m_{1}-m_{2}}{S}\times 24$$
where *m*_1_ was the test cup mass before the test; *m*_2_ was the test cup mass after the test; *S* was the test cup mouth area; and the obtained *WVT* rate was 9.6 Kgm^−2^d^−1^, which could confirm that the all-electrospun textile film had excellent air permeability. In other words, electrospun nanofibers had excellent wearing comfort, ensuring human thermal comfort and wide personal practical applications after their integration into wearable electronic devices.

### 3.5. Test of Reliability and Application of TENGs

To explore the durability and stability of TENGs, the fabric current outputted after bending the two friction layers for 1000 cycles was tested and compared. It could be clearly seen that there was no significant mechanical damage to the surface, shown in Figure 7b. The outputted current did not present a significant reduction. It was indicated that the nanofiber films spun prepared had good mechanical properties.

To accurately evaluate the practical application of TENGs, a series of tests was carried out and the combined light chain composed of 110 LEDs in series could be lit by using the TENGs (Figure 8a), which showed that it was fully capable of outputting power to meet some low-power compatible electronic devices and it had the advantage of providing reliable power supply for wearable systems.

To demonstrate the charging capability of TENGs, a circuit composed of a commercial capacitor and bridge rectifier was designed to test its charging efficiency. The charge in the capacitors (1 μF, 10 μF, 47 μF) to 1.2V with 4 s, 36 s, 180 s, respectively, is shown in Figure 8b.

To further demonstrate the application of TENGs based on all-electrospun nanofilms, they could be easily combined with textiles to harvest the mechanical energy generated from the human body, As shown in Figure 8c,d, two samples were attached to the clothing. The movement of the human body will lead to the contact and separation of the two friction layers, thereby converting the mechanical energy of the human body into electrical energy. In these experiments, the highest open-circuit voltages could reach 60 V and 40 V. The results showed that it was fully capable of collecting and supplying energy in daily life, and it not only functions as a power source, but also had the ability to collect external stimuli to generate signals. In the future, it may have the ability to collect human body posture, pulse, heartbeat and other human health information while supplying power.

## 4. Conclusions

In this work, an electrospun PA66/graphene nanofiber film with a secondary nano-network structure, as a tribo-cathode layer, was fabricated, and it was assembled with the electrospun PVDF film as the tribo-anode layer to form all-electrospun film-based TENGs. The rough surface of the two friction layers not only increased the effective contact area, but also improved the breathability. Furthermore, the porous fiber network in each functional layer guaranteed the air permeability of the TENGs. Compared with the pure PA66 fiber film, the introduced graphene increased the effective contact area of the PA66 nanofiber film and graphene could act as a charge trapping site, enhancing the electrical properties of the TENGs. The fabricated TENGs exhibited high Voc, Isc and peak power density of 180 V, 7.8 µA and 2.67 W/m^2^, respectively. In addition, TENGs maintained excellent performance after 1000 cycles under cyclic pressure, keeping its advantages in durability and stability. TENGs with good comfort and high flexibility could obtain biomechanical energy from various human motions. There are great prospects in the system integration of wearable electronic devices.

## Figures and Tables

**Figure 1 materials-15-05191-f001:**
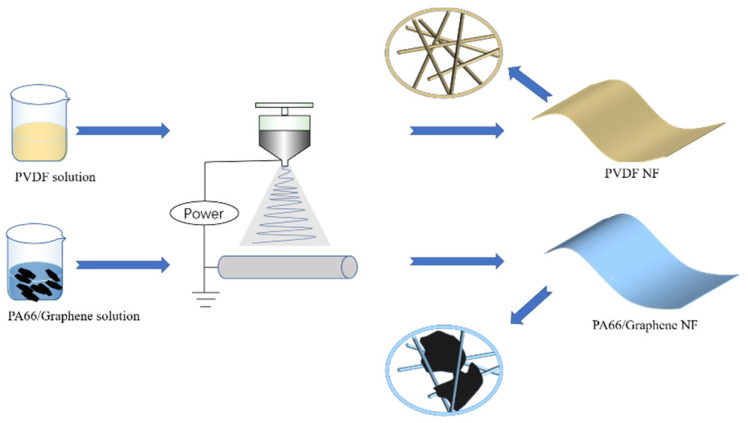
Schematic diagram of electrospinning film.

**Figure 2 materials-15-05191-f002:**
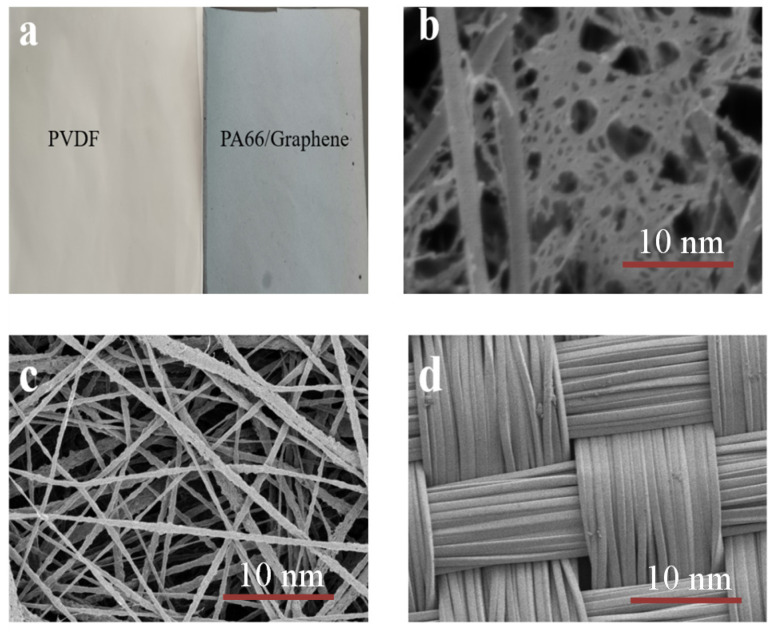
(**a**) Electrospun PVDF and PA66/graphene films; (**b**) SEM image of PA66/graphene film; (**c**) SEM image of PVDF film; (**d**) SEM image of plain weave conductive fabric.

**Figure 3 materials-15-05191-f003:**
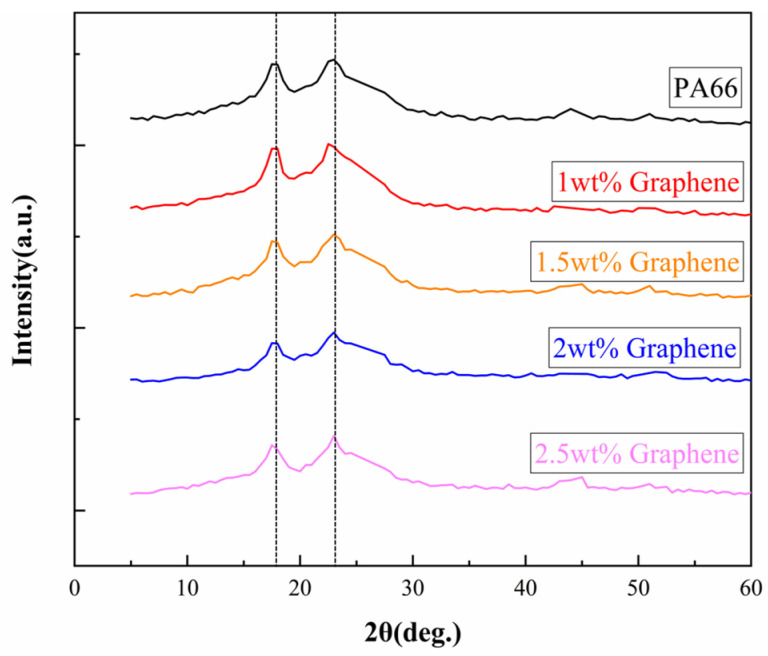
XRD patterns of PA66/graphene with different graphene mass fractions.

**Figure 4 materials-15-05191-f004:**
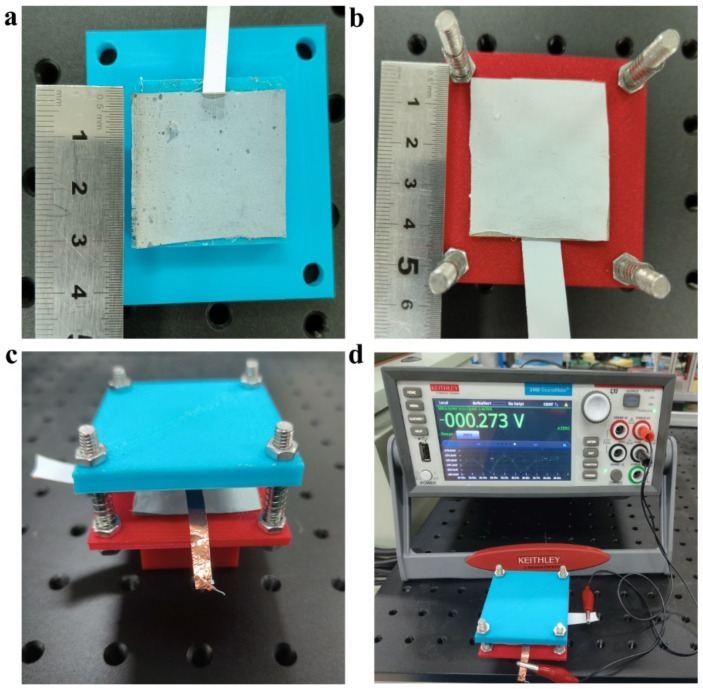
Testing flow diagram of the contact-separated TENGs. (**a**) PA66/graphene pasted on the upper plate; (**b**) PVDF film pasted on the bottom plate; (**c**) installation diagram after assembly; (**d**) signal testing diagram.

**Figure 5 materials-15-05191-f005:**
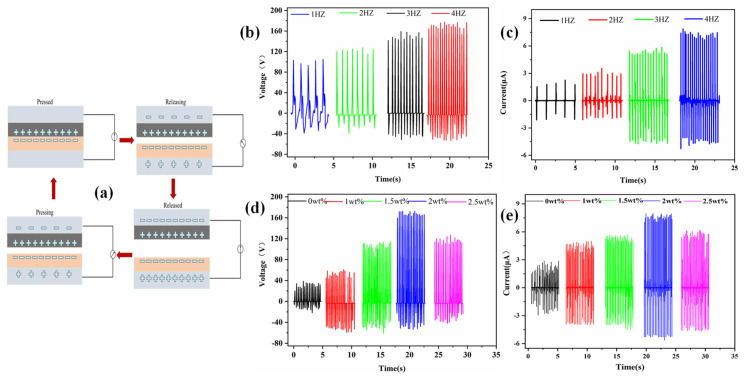
Output performance of nanofiber-film-based TENGs. (**a**) Working principle of the TENGs; (**b**) open-circuit voltage outputted of TENGs at different frequencies; (**c**) short-circuit current output of TENGs at different frequencies; (**d**,**e**) effects of the thickness of PA66/graphene.

**Figure 6 materials-15-05191-f006:**
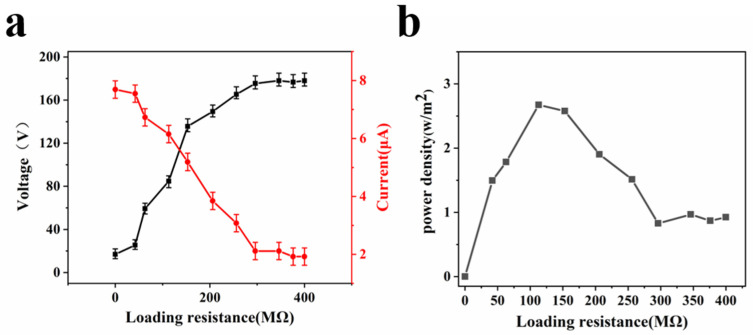
Relationship of load resistance on output of voltage and current. (**a**) Relationship of load resistance on output of voltage; (**b**) relationship of load resistance on output of current.

**Figure 7 materials-15-05191-f007:**
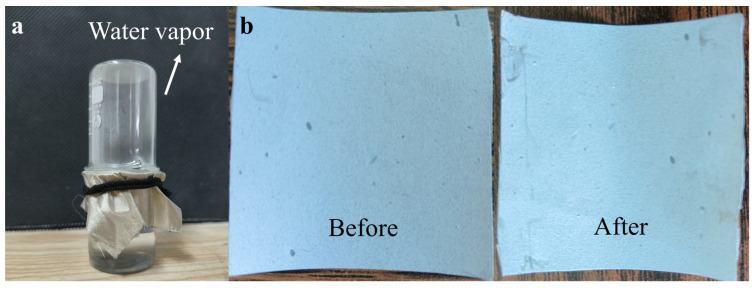
(**a**) Water vapor transmission rate test (WVT); (**b**) comparison before and after a thousand times of bending.

**Figure 8 materials-15-05191-f008:**
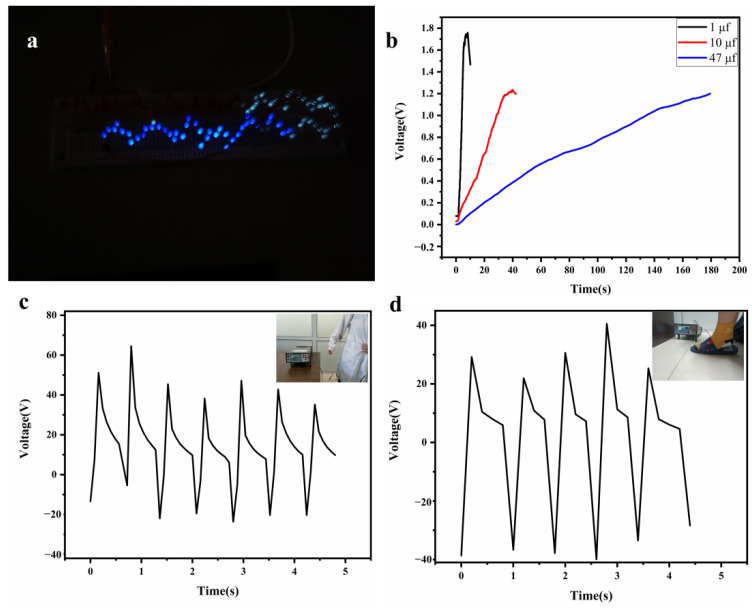
(**a**) 110 commercial led tandem light chains lit by TENGs; (**b**) three commercial capacitors to 1.2 V, respectively, charged by TENGs; (**c**) mechanical energy of the swing arm movement collected by TENGs; (**d**) mechanical energy from walking motions collected by TENGs.

## Data Availability

The data that support the findings of this study have not been made available but can be obtained from the author upon request.

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
