# Peer review of "Electrospun PA66/Graphene Fiber Films and Application on Flexible Triboelectric Nanogenerators"

_materials, 2022, doi:10.3390/ma15155191_

Round 1

Reviewer 1 Report

Dear authors

The manuscript is interesting, has a clear purpose, and has several results. I have two questions:

Q1 – What is the thickness of the layers?

Q2 – Why to use a double-side conductive tape if you have a conductive fabric? How does it affect the mechanical response of the system?

Q3 – Did you perform a dielectric/piezoelectric constant measurements? 

Q4 – Did the PVDF was electrical pooled? Does it affect the efficiency of the system?

Q5 – Can the water affects the TENG since it changes the electrical conductivity? Did you perform the experiments with a soak TENG?

Author Response

Dear reviewer,

  Thank you for your work.  The response is presented as attachments. If you have any problems, please do not contact me. Thank you again.
Best wishes,
Dr. Yu

Reviewer 2 Report

Dear Authors,

First of all I appreciate the idea of presented study but, in my opinion, some some issues must be addressed before publication:

1. please present the experimental setup used to obtain data presented in figure 4, figure 5 and figure 7. What is the reference used in these tests?

2. page 4 lines 161-170: in my opinion this paragraph is fuzzy. There are explanations, which are not consistent with the images presented in figure 2. I don't understand correlations between triboelectric properties and SEM images. Please give more information or present it in way to make the explanation clearer.

3. page 5 line 177-179: how can you describe complete structure? How a XRD diffraction pattern is related with this? Please modify if it is a typo.

4. page 9 line 303-305: Please modify or remove the final phrase. It is to general. Although empowering, and I appreciate the enthusiasm ("Star Trek style"), I think that you should stay to the subject, for now.

Author Response

(The authors gave the same response as above.)

Reviewer 3 Report

Please, use one style of capitals letters in the name “PA66”.

Please, support this part of the sentence with reference “…which could effectively increase the friction performance and air permeability [DOI: 10.3390/MA14061428]….“

Line 152: „…turboelectric…“ . May be „triboelectric“?

Figure 3: What is the origin of small peaks around 45 and 50 degrees?

Please, improve quality of Figure 5.

The voltage values seem to be really good, could you please add the video of experiment to supplementary materials just to make the readers confident in functionality of these materials.

Author Response

(The authors gave the same response as above.)
